# Prognostic Value of SDF-1α Expression in Patients with Esophageal Squamous Cell Carcinoma Receiving Esophagectomy

**DOI:** 10.3390/cancers12051067

**Published:** 2020-04-25

**Authors:** Yen-Hao Chen, Shau-Hsuan Li, Hung-I Lu, Chien-Ming Lo

**Affiliations:** 1Department of Hematology—Oncology, Kaohsiung Chang Gung Memorial Hospital and Chang Gung University College of Medicine, Kaohsiung 833, Taiwan; lee0624@cgmh.org.tw; 2Graduate Institute of Clinical Medical Sciences, College of Medicine, Chang Gung University, Taoyuan 333, Taiwan; 3School of Medicine, Chung Shan Medical University, Taichung 402, Taiwan; 4Department of Thoracic & Cardiovascular Surgery, Kaohsiung Chang Gung Memorial Hospital and Chang Gung University College of Medicine, Kaohsiung 833, Taiwan; luhungi@cgmh.org.tw (H.-I.L.); t123207424@cgmh.org.tw (C.-M.L.)

**Keywords:** SDF-1α, esophageal cancer, squamous cell carcinoma, esophagectomy

## Abstract

Stromal cell-derived factor-1α (SDF-1α) is a chemokine that has been reported to be involved in tumor progression in several malignancies. This study aimed to evaluate the crucial role of SDF-1α in patients with esophageal squamous cell carcinoma (ESCC) who underwent esophagectomy. A total of 169 patients with ESCC were identified, including overexpression of SDF-1α in 60 patients and low expression of SDF-1α in 109 patients by immunohistochemical analysis. Two ESCC cell lines, TE1 and KYSE30, were selected to evaluate the tumor cell proliferative effect of SDF-1α. Univariate and multivariate analyses showed that high tumor (T) status, positive lymph node metastasis, tumors located in the upper esophagus, and SDF-1α overexpression were significantly related to worse disease-free survival and overall survival. In addition, the two cell lines were treated with SDF-1α, AMD3100 (an SDF-1α-ligand receptor antagonist), and chemotherapeutic agents (cisplatin). Our in vitro study results showed that SDF-1α promoted the proliferation of tumor cells, and blocking the SDF-1α pathway displayed a growth inhibition effect in a dose-dependent manner. SDF-1α plays an important role in the progression of ESCC and is an independent prognostic factor for ESCC patients who underwent esophagectomy.

## 1. Introduction

Esophageal squamous cell carcinoma (ESCC) is known to be among the most aggressive cancers. In Taiwan, ESCC accounts for 90% of esophageal cancer and is one of the top reasons for cancer-related deaths [1]. For early-stage ESCC, radical esophagectomy is the gold-standard treatment; however, in spite of significant developments in medical techniques, the 5-year survival rate for patients with ESCC who underwent esophagectomy is still poor [1,2,3]. Thus, the exploration of novel molecular prognostic models for tumor progression in ESCC is an important research priority.

Stromal cell-derived factor-1α (SDF-1α) is a homeostatic chemokine that is constitutively expressed in several organs, and SDF-1α secretion is related to tissue damage. C-X-C chemokine receptor type 4 (CXCR4) is a receptor of SDF-1α, and the binding of SDF-1α to CXCR4 modulates downstream signaling pathways, resulting in a variety of responses, such as chemotaxis, tumor cell proliferation, distant metastasis, and gene transcription [4,5].

Growing evidence has shown that the SDF-1α/CXCR4 axis could regulate the activities of tumor cells and play a key part in modulating the interaction of CXCR4+ tumor cells and SDF-1α-expressing organs. Albert et al. showed that overexpression of SDF-1α/CXCR4 appears to activate cellular functions, including migration, invasion, and metastasis in head and neck squamous cell carcinoma [6]. The SDF-1α/CXCR4 axis is associated with tumor aggressiveness, high metastatic potential, and poor prognosis. Wang et al. revealed that SDF-1α is highly expressed in lung cancer cells and is related to distant metastasis [7]. In addition, the SDF-1α/CXCR4 axis modulates tumor stem cells to initiate and promote lung cancer. Another study, reported by Amara, demonstrated that SDF-1/CXCR4 enhances liver metastases, causing poor outcomes in patients with colorectal cancer [8].

Recently, the role of the SDF-1α/CXCR4 axis has been explored in some studies [9,10]. Lin et al. revealed that SDF-1α-induced esophageal cancer cell invasion was inhibited by suppressing the downstream signal pathways through the inhibition of ras-related C3 botulinum toxin substrate 1 activity [9]. Wang et al. showed that there was a significant difference in 5-year survival rates between ESCC patients with positive and negative SDF-1α, while the CXCR4-positive group had better 5-year survival rates compared to those with CXCR4- negative [10]. To the best of our knowledge, the role of SDF-1α in ESCC tumor progression is limited. The objective of the current study was to explore the role of SDF-1α in determining the clinical outcomes of ESCC patients who received esophagectomy and the tumor cell proliferative effect of SDF-1α in an in vitro study.

## 2. Results

### 2.1. Patient Selection

The ESCC database was retrospectively reviewed. First, patients with R1 or R2 resection, carcinoma in situ, distant metastasis, or a history of second primary malignancy were excluded, and only those ESCC patients who received radical esophagectomy as treatment were included. In addition, patients who received preoperative chemotherapy, radiotherapy or concurrent chemoradiotherapy before esophagectomy, or palliative surgical resection, were excluded. Adjuvant treatment, such as radiation alone or concurrent chemoradiotherapy, was administered to patients with adverse pathologic features. Finally, a total of 169 ESCC patients who received radical esophagectomy and who met the criteria were identified. There were 163 men and six women, with a mean age of 55 years (range: 29–80 years). Most ESCC patients who received esophagectomy were early-stage, including higher percentage T1–2, negative nodal metastasis, and stage I–II. More than half of ESCC patients had tumors located in the middle or lower third of the esophagus. At the time of analysis, the median period of follow-up was 67.5 months for the 60 survivors and 36.0 months for all 169 patients; the median disease-free survival (DFS) and overall survival (OS) were 25.2 months and 36.0 months, respectively. The clinicopathological characteristics of these patients are shown in Table 1.

### 2.2. Clinical Outcomes of ESCC Patients Who Received Esophagectomy

The expression of SDF-1α in the immunohistochemical staining is shown in Figure 1. Among the 169 patients, 60 patients (35.5%) were classified as having an overexpression of SDF-1α, and 109 patients (64.5%) had a low expression of SDF-1α. The baseline characteristics did not differ significantly between these two groups, including age, tumor location, and tumor grade; however, there were higher percentages of advanced tumor stage and adjuvant treatment in the SDF-1α overexpression group (Table 2).

With respect to the DFS, the univariate analysis revealed no significant differences in sex, tumor location, and tumor grade. The 63 patients aged below 60-years-old were found to have better DFS than the other 106 patients, aged over 60 years old (39.4 months vs. 14.8 months, *p* = 0.007). Superior DFS was mentioned in the 90 patients with T1–T2 status in comparison with the 79 patients with T3–T4 status (56.4 months vs. 9.8 months, *p* < 0.001), and higher DFS was found in the 116 patients with negative nodal metastasis compared to the 53 patients with positive nodal metastasis (45.7 months vs. 7.1 months, *p* < 0.001). Moreover, the 114 stage I–II ESCC patients had better DFS than the 55 stage III–IVA ESCC patients (45.7 months vs. 6.8 months, *p* < 0.001). Worse DFS was mentioned in the 99 patients who received adjuvant treatment in comparison with the other 70 patients who did not (10.3 months vs. 65.9 months, *p* < 0.001). The 109 patients with low expression of SDF-1α were found to have significantly longer DFS compared to the 60 patients with overexpression of SDF-1α (50.0 months vs. 6.9 months, *p* < 0.001, Figure 2A). The multivariable analysis revealed that T1–T2 (*p* < 0.001), negative N status (*p* = 0.002), ESCC located in the middle and lower esophagus (*p* = 0.006), and low expression of SDF-1α (*p* < 0.001) were independent predictive factors for better DFS.

The univariate analysis of OS showed that sex and tumor grade were not prognostic factors of OS. Meanwhile, 63 patients aged below 60 years old had higher OS compared to the 106 patients aged 60 years or older (64.2 months vs. 23.7 months, *p* = 0.009). The longer OS was mentioned in 90 T1–T2 patients in comparison with the 79 T3–T4 patients (67.2 months vs. 15.1 months, *p* < 0.001). The 116 patients with negative nodal metastasis had significantly improved OS than in the 53 patients with positive nodal metastasis (65.6 months vs. 13.0 months, *p* < 0.001). Moreover, superior OS was found in the 114 stage I–II ESCC patients, who were found to have better OS with stage I–II ESCC than the 55 stage III–IVA ESCC patients (65.8 months vs. 12.0 months, *p* < 0.001). Superior OS was mentioned in the 141 middle/lower third ESCC patients compared to the 28 lower third ESCC patients (45.0 months vs. 16.7 months, *p* = 0.035). The 99 patients who experienced adjuvant treatment had worse OS than the other 70 patients who did not (15.5 months vs. 69.3 months, *p* < 0.001). Better OS was found in the 109 patients with low expression of SDF-1α than the other 60 patients with overexpression of SDF-1α (67.5 months vs. 10.0 months, *p* < 0.001, Figure 2B). In addition, T1–T2 status (*p* = 0.001), negative N status (*p* = 0.001), tumors located in the middle and lower esophagus (*p* = 0.002), and low expression of SDF-1α (*p* < 0.001) were independent prognostic predictors of better OS in a multivariate analysis. The univariate and multivariate analyses of DFS and OS in 90 ESCC patients are demonstrated in Table 3 and Table 4, respectively.

### 2.3. SDF-1α Promotes Tumor Cell Proliferation In Vitro

In this study, two ESCC cell lines, TE1 and KYSE30, were used to test the role of SDF-1α in tumor cell proliferation. The cell lines were first treated with chemotherapeutic agents (cisplatin), SDF-1α, and AMD3100 (a CXCR4 antagonist), to determine the dependence of tumor proliferation on SDF-1α. The results suggest that SDF-1α could promote the proliferation of tumor cells and that AMD3100 could inhibit tumor cell proliferation in a dose-dependent reduction at 48 h (Figure 3). Furthermore, the TE1 and KYSE30 were treated with SDF-1α combined with chemotherapeutic agents (cisplatin) or AMD3100, and the results demonstrate that the SDF-1α/CXCR4 axis suppresses ESCC cell growth (Figure 4).

## 3. Discussion

SDF-1α, also called as C-X-C motif chemokine 12 (CXCL12), is a homeostatic chemokine that has recently attracted much attention in the immune system, inflammation/infection, nervous system, tissue damage, and hematopoiesis research. CXCR4, a G-protein-coupled receptor, is a receptor for SDF-1α and a crucial mediator of cell migration in both leukocytes and tumor cells. Several studies suggested that the SDF-1α/CXCR4 axis is associated with several biological processes, such as immune response, hematopoiesis, cardiovascular system organogenesis, and cancer progression. Moreover, SDF-1α/CXCR4 signaling plays a key role in common malignancies and has modulated a lot of responses, such as chemotaxis, tumor cell proliferation, invasion, and metastasis [4,5]. Otsuka et al. reported that the SDF-1α/CXCR4 axis modulates chemotaxis, tumor migration, and angiogenesis by activating several signaling pathways, and an in vivo study also demonstrated that blocking the SDF-1α/CXCR4 axis results in a significant reduction in tumor cell progression in non-small cell lung cancer [11]. In pancreatic cancer, CXCR4 overexpression is a prognostic factor of poor OS; CXCR4 blocking significantly mediates the phenotype of pancreatic cancer cells, inhibiting tumor cell proliferation, invasion, and metastasis [12]. Growing evidence has confirmed that the activation of SDF-1α/CXCR4 signaling promotes tumor cell proliferation, migration, survival, gene transcription, and blockade of the SDF-1α/CXCR4 axis reverses this phenomenon and results in cell cycle arrest, apoptosis, and the inhibition of downstream signaling [6,13,14,15].

Several studies have shown that the level of SDF-1α expression is related to disease severity in ESCC patients, such as tumor size, lymph node metastasis, and survival outcome [16,17]. Sasaki et al. reported that positive SDF-1α expression was significantly correlated with lymph node metastasis and lymphatic invasion [16,17]. Moreover, worse DFS and OS were noted in ESCC patients with overexpression of SDF-1α compared to those with low expression of SDF-1α. However, the expression of CXCR4 revealed no correlation with clinicopathological variables and prognosis in this study. A Japanese study showed that positive SDF-1α expression is related to lower recurrence-free survival rates, and SDF-1α expression is an independent predictive factor for recurrence in ESCC patients [16,17]. In addition, SDF-1α overexpression enhanced ESCC tumor proliferation and AMD3100 decreased tumor cell growth irrespective of SDF-1α expression in vitro; AMD3100 also significantly decreased ESCC tumor size in an animal model. Our study showed that SDF-1α was related to DFS and OS, and SDF-1α overexpression was an independent prognostic predictor for worse DFS and OS in ESCC patients. In addition, SDF-1α-promoting tumor cell proliferation and AMD3100-inhibiting tumor cell growth irrespective of SDF-1α were also demonstrated in our in vitro study.

There are several limitations to our study. First, the study was retrospectively designed, and all patients were from a single hospital, so a relatively small sample size was used. Second, only six female patients were enrolled in our study, so there may be a bias in the analysis of survival outcomes between male and female patients. However, our study confirms the role of SDF-1α in ESCC. Further studies in a larger population and animal studies are needed to validate the findings of our study.

## 4. Materials and Methods

### 4.1. Patient Selection

Between January 2001 and December 2015, we retrospectively reviewed patients with ESCC who underwent esophagectomy at Kaohsiung Chang Gung Memorial Hospital. First, we excluded patients who had a second primary malignancy. In addition, patients who received other modalities before esophagectomy were also excluded, such as preoperative chemotherapy, radiotherapy, or chemoradiotherapy. Finally, we identified a total of 169 patients. Chest computed tomography, endoscopic ultrasonography, and positron emission tomography scans were performed to determine the clinical stage and the possibility of surgical resection for each patient, and pathology staging was performed according to the system outlined in the 8th edition of the American Joint Committee on Cancer staging system [18].

### 4.2. Immunohistochemical Analysis

Immunohistochemical staining was performed using the immunoperoxidase technique. Slides were prepared from formalin-fixed, paraffin-embedded tissue sections by cutting at 4 μm, and staining was performed using primary antibodies against SDF-1α (MAB350, 15 µg/mL; R&D Systems, Minneapolis, MN, USA). First, we performed deparaffinization and rehydration, and then we incubated the slides in heat-induced epitope retrieval for 20 min using a hot water bath set to 95 °C. Immunodetection was performed using an LSAB2 kit (Dako, Carpinteria, CA, USA). Some 3-3’-diaminobenzidine was used for color development, and hematoxylin was used for counterstaining. Human bile duct epithelial cells were used as positive controls, and incubation without the primary antibody was used as a negative control. Staining was independently assessed by two pathologists (Chao-Cheng Huang and Wan-Ting Huang) blinded to the clinicopathological features and prognosis (Appendix A). The expression of SDF-1α was assessed according to a semi-quantitative immunoreactive score (IRS) [19]. The IRS was obtained by multiplying the staining intensity with a range of 0–12, including a score of 0 (no positively stained cells), 1 (*<*10% of positively stained cells), 2 (10–50% of positively stained cells), 3 (51–80% of positively stained cells) and 4 (*≥*81% of positively stained cells), and the expression of the histological grade (0: no staining, 1: weak, 2: moderate and 3: strong staining). An IRS score of 6–12 was considered as overexpression.

### 4.3. Cell Culture and MTT Assay

The ESCC cell lines TE1 and KYSE30 have been established. KYSE30 cells were obtained from Public Health England (London, UK), and TE1 cells were purchased from the Cell Resource Center for Biomedical Research Institute of Development, Aging and Cancer (Tohoku University, Sendai, Japan). Cells were cultured in RPMI 1640 or PRMI 1640/F12 (1:1) media with 5% fetal bovine serum, 100 U/mL penicillin, 100 μg/mL streptomycin, 0.25 μg/mL amphotericin B, and 2 mmol/L L-glutamine. To examine the role of SDF-1α in determining the malignant properties of ESCC cells, we treated these ESCC cell lines with chemotherapeutic agents (cisplatin), SDF-1α (MAB350, 100 ng/mL, R&D Systems), and AMD3100 (Sigma-Aldrich, Saint Louis, MO, USA). Each cell line (2500 cells) was incubated in a 200 μL solution containing SDF-1α, cisplatin, or AMD3100 at different concentrations in triplicate in a 96-well flat-bottomed plate. An MTT (3-[4-5-dimenthylthiazol-2-yl]-2,5-diphenyltetrazolium bromide; 0.5 mg/mL; Sigma-Aldrich) assay was performed to explore the cell proliferative activity of SDF-1α in these ESCC cell lines. Each cell line (7000 cells) was incubated, along with control cells, in triplicate in a 96-well flat-bottomed plate. After incubation for 96 h at 37 °C, 100 μL of MTT was added to each well, and the cells were incubated for 4 h. The supernatant was discarded, and the crystalline products were eluted with DMSO (50 μL/well, Sigma). A colorimetric evaluation was conducted using a spectrophotometer at 570 nm.

### 4.4. Statistical Analysis

The data were analyzed using the SPSS 19 software package (IBM, Armonk, NY, USA). The Student’s *t*-test and Fisher’s exact test or a Chi-squared test were used to compare the differences of categorical variables between groups. DFS was calculated from the date of esophagectomy to the date of recurrence of tumor or death, and OS was defined as the time from the date of esophageal cancer diagnosis to the date of death from all causes or until the end of the last follow-up. Kaplan–Meier curves were constructed to estimate DFS and OS, and the *p*-values were determined by the log-rank test. For multivariate analyses, a Cox regression was performed, calculating the HR with a 95% CI. All prognostic factors with *p*-values < 0.1 in the univariate model were further entered into the multivariate analysis. All the statistical tests were two-sided, and *p* < 0.05 was used to indicate a statistically significant difference.

### 4.5. Ethics Statement

The study was conducted in accordance with the Declaration of Helsinki and approved by the Chang Gung Medical Foundation Institutional Review Board (201701862B0 and 201801383B0). Written informed consent was waived due to the retrospective design.

## 5. Conclusions

Our study suggests that SDF-1α plays an important role in the progression of ESCC and is an independent prognostic factor for ESCC patients who underwent esophagectomy. Further larger studies are warranted to validate the results of our study.

## Figures and Tables

**Figure 1 cancers-12-01067-f001:**
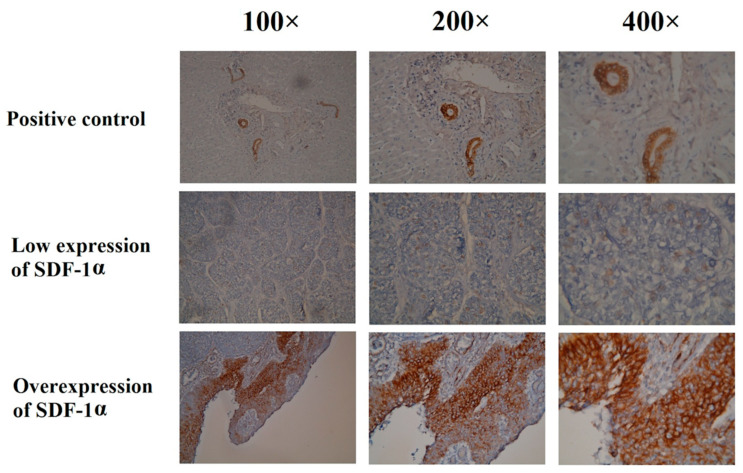
Immunohistochemical staining of the SDF-1α in esophageal squamous cell carcinoma patients undergoing esophagectomy.

**Figure 2 cancers-12-01067-f002:**
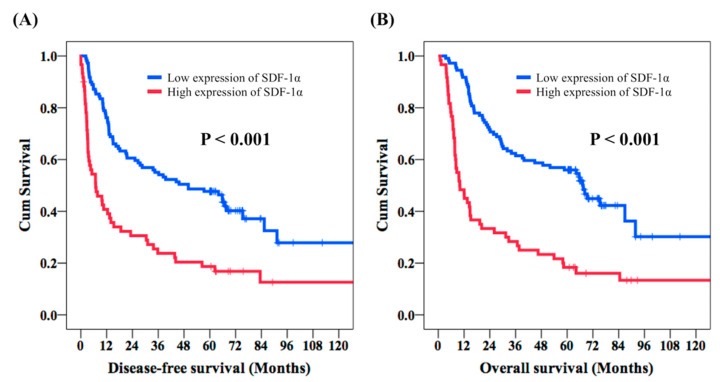
Comparison of disease-free survival (DFS) and overall survival (OS) in esophageal squamous cell carcinoma patients who underwent esophagectomy according to the expression of SDF-1α. (**A**) DFS and (**B**) OS.

**Figure 3 cancers-12-01067-f003:**
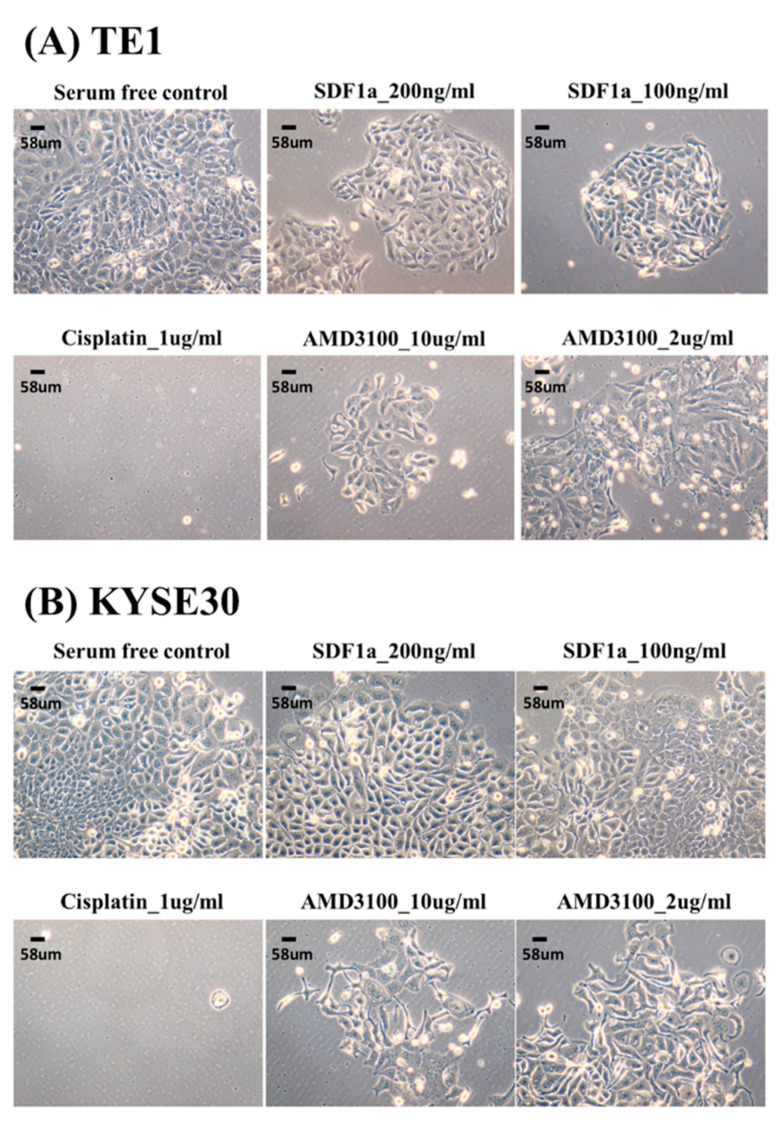
The effect of tumor cell proliferation of SDF-1α, AMD3100, and chemotherapy in esophageal squamous cell carcinoma cell lines, TE1, and KYSE30.

**Figure 4 cancers-12-01067-f004:**
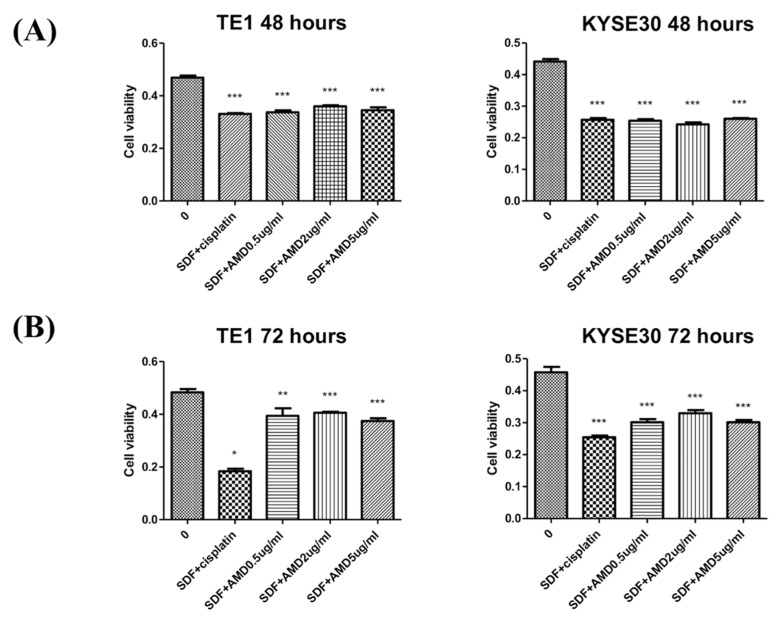
SDF-1α/CXCR4 axis inhibits tumor cell proliferation in a dose-dependent reduction on the two esophageal squamous cell carcinoma cell lines, TE1 and KYSE30. (**A**) 48 h. (**B**) 72 h. Columns, mean; bars, standard deviation. Significant difference from the indicated comparisons: * *p* < 0.05, ** *p* < 0.01, and *** *p* < 0.001.

**Table 1 cancers-12-01067-t001:** Characteristics of 169 patients with esophageal squamous cell carcinoma receiving esophagectomy.

Characteristics	Patient Numbers (%)
Age (years)	55 years old (29–80)
Sex	
Male	163 (96.4%)
Female	6 (3.6%)
pT status	
1	56 (33.1%)
2	34 (20.1%)
3	63 (37.3%)
4	16 (9.5%)
pN status	
0	116 (68.6%)
1	33 (19.6%)
2	13 (7.7%)
3	7 (4.1%)
Tumor stage	
I	51 (30.2%)
II	63 (37.3%)
III	34 (20.1%)
IVA	21 (12.4%)
Location	
Upper	28 (16.6%)
Middle	63 (37.3%)
Lower	78 (46.1%)
Grade	
1	17 (10.1%)
2	108 (63.9%)
3	44 (26.0%)

**Table 2 cancers-12-01067-t002:** Comparison of clinicopathological parameters in 169 patients with esophageal squamous cell carcinoma receiving esophagectomy.

Characteristics	Overexpression of SDF-1α (*N* = 60)	Low Expression of SDF-1α (*N* = 109)	*p* Value
Age			
<60 years	37 (61.7%)	69 (63.3%)	0.87
≥ 60 years	23 (38.3%)	40 (36.7%)	
Tumor stage			
I + II	27 (45.0%)	87 (79.8%)	<0.001 *
III + IVA	33 (55.0%)	22 (20.2%)	
Location			
Upper	9 (15.0%)	19 (17.4%)	0.83
Middle + Lower	51 (85.0%)	90 (82.6%)	
Grade			
1 + 2	40 (66.7%)	85 (78.0%)	0.14
3	20 (33.3%)	24 (22.0%)	
Adjuvant treatment			
Yes	49 (81.7%)	50 (45.9%)	<0.001 *
No	11 (18.3%)	59 (54.1%)	

* Statistically significant.

**Table 3 cancers-12-01067-t003:** Univariate and multivariate analysis of disease-free survival (DFS) in 169 patients with esophageal squamous cell carcinoma receiving esophagectomy.

Parameters	Number of Patients	DFS (Months)	Univariate Analysis	Multivariate Analysis	
			*p*-Value	HR (95% CI)	*p*-Value
Age			0.007 *		
<60 years	63 (37.3%)	39.4			
≥60 years	106 (62.7%)	14.8			
Sex			0.14		
Male	163 (96.4%)	21.5			
Female	6 (3.6%)	NR			
pT status			<0.001 *		
1 + 2	90 (53.3%)	56.4		0.50 (0.34–0.73)	<0.001 *
3 + 4	79 (46.7%)	9.8			
pN status			<0.001 *		
0	116 (68.6%)	45.7		0.53 (0.36–0.79)	0.002 *
1 + 2 + 3	53 (31.4%)	7.1			
Tumor stage			<0.001 *		
I + II	114 (67.5%)	45.7			
III + IVA	55 (32.5%)	6.8			
Location			0.1		
Upper	28 (16.6%)	13.7		0.51 (0.31–0.82)	0.006 *
Middle + Lower	141 (83.4%)	33.8			
Grade			0.16		
1 + 2	125 (74.0%)	31.1			
3	44 (26.0%)	13.1			
Adjuvant treatment			<0.001 *		
Yes	99 (58.6%)	10.3			
No	70 (41.4%)	65.9			
SDF-1α expression			<0.001 *		
High	60 (35.5%)	6.9			
Low	109 (64.5%)	50		0.49 (0.33–0.72)	<0.001 *

NR: not reach; HR: hazard ratio; CI: confidence interval * Statistically significant.

**Table 4 cancers-12-01067-t004:** Univariate and multivariate analysis of overall survival (OS) in 169 patients with esophageal squamous cell carcinoma who received esophagectomy.

Parameters	Number of Patients	Univariate Analysis		Multivariate Analysis	
		OS (Months)	*p*-Value	HR (95% CI)	*p*-Value
Age			0.009 *		
<60 years	63 (37.3%)	64.2			
≥60 years	106 (62.7%)	23.7			
Sex			0.2		
Male	163 (96.4%)	32.8			
Female	6 (3.6%)	NR			
pT status			<0.001 *		
1 + 2	90 (53.3%)	67.2		0.50 (0.34−0.75)	0.001 *
3 + 4	79 (46.7%)	15.1			
pN status			<0.001 *		
0	116 (68.6%)	65.6		0.51 (0.34−0.76)	0.001 *
1 + 2 + 3	53 (31.4%)	13			
Tumor stage			<0.001 *		
I + II	114 (67.5%)	65.8			
III + IVA	55 (32.5%)	12			
Location			0.035 *		
Upper	28 (16.6%)	16.7		0.46 (0.29−0.75)	0.002 *
Middle + Lower	141 (83.4%)	45			
Grade			0.07		
1 + 2	125 (74.0%)	48.7			
3	44 (26.0%)	22.4			
Adjuvant treatment			<0.001 *		
Yes	99 (58.6%)	15.5			
No	70 (41.4%)	69.3			
SDF-1α expression			<0.001 *		
High	60 (35.5%)	10			
Low	109 (64.5%)	67.5		0.40 (0.27-0.60)	<0.001 *

NR: not reach; HR: hazard ratio; CI: confidence interval * Statistically significant.

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
