# Peer review of "Prognostic Value of SDF-1α Expression in Patients with Esophageal Squamous Cell Carcinoma Receiving Esophagectomy"

_cancers, 2020, doi:10.3390/cancers12051067_

Round 1
Reviewer 1 Report
General Comments
General:
In this paper, the authors evaluated the crucial role of SDF-1α in patients with esophageal squamous cell carcinoma (ESCC) who underwent esophagectomy. In this work, immunohistochemical staining was performed to evaluate the crucial role of SDF-1α in patients with esophageal squamous cell carcinoma (ESCC) who underwent esophagectomy. Moreover, TE1 and KYSE30, were selected to evaluate the tumor cell proliferative effect of SDF-1α. The results indicated that SDF-1α plays an important role in the progression of ESCC and is an independent prognostic factor for ESCC patients who underwent esophagectomy. The novelty of the manuscript is limited and a little uncompleted. Only 4 figures and 3 tables are revealed in the manuscript. The reviewer has some suggestions as follows:
Specific comments
- Please list more references of the expression of CXCR4, SDF-1α and its correlation to prognosis in ESCC. (Ai Zheng. 2009 Feb;28(2):154-8; Mol Carcinog. 2014 May;53(5):360-79….etc)
- Please provide the scales in Fig. 2.
- It should be better if the authors provide some western blotting results to clarify the relation between the level of SDF-1α expression and ESCC.
- It should be better if the authors provide some flow cytometry results to clarify the relation between SDF-1α and proliferation.
- Please provide the IRB license number of clinical trials.
- Please provide the criteria to assess the staining results. For example, how to define the 50% of positively stained cells? (the size of field of view? using which automatic software or manual control?)
Reviewer 2 Report
Considering the type of disease, the present work is potentially interesting and innovative. Mainly due to the retrospective nature and the heterogeneity of patients, some critical points need to be clarified:
- The number of patients is very good for the type of disease and in a single institution. The authors should report the median follow-up, the median disease-free survival and the median overall survival for the entire cohort of patients in section 2.1
- Please specify if all patients underwent radical resection or not; if not, please add in Table 1 and perform univariate (and eventually multivariate) analysis for DFS and OS also for this baseline characteristic
- Please describe any post-operative treatment that could have influenced outcome especially in more advanced stages and perform univariate (and eventually multivariate) analysis for DFS and OS also for this parameter
- Since IHC results are not well-established for SDF-1α expression, a Supplementary Table reporting detailed results for expression should be added and the concordance among pathologists' assessment in IHC results should be reported in the text
- It would be good to look at if there were differences in age, stage, grade, location, R0 resection rate, types of post-operative treatment between the SDF-1α + and - groups to ensure there were no other biases that influenced the prognostic significance of SDF-1α expression
- In methods - statistical analyses, more details regarding univariate and multivariate analysis tests should be given
- References should be added in the manuscript
Reviewer 3 Report
This is an interesting manuscript regarding the SDF-1α/CXCR4 axis role on esophageal ca. Hope this could add to a growing body of literature of SDF-1α on the prognostication of esophageal carcinoma aiding in the future treatment of this highly poor prognosis cancer.
I have only one issue need to be address. At present, Is there any blocking agent against SDF-1α overexpression ?
Round 2
Reviewer 1 Report
In the paper entitled "Prognostic Value of SDF-1α Expression in Patients with Esophageal Squamous Cell Carcinoma Receiving Esophagectomy", the authors answered and added some contents according to reviewer’s suggestions on a point-by-point basis. In view of the scope, novelty, and quality of this work, I would recommend this paper to be published in ultrasound in Cancers.
Reviewer 2 Report
All the requested clarifications have been provided in the revised version of the article. I have no other comments to add.